# Development and validation of a nomogram for predicting false negative IGRA results in pulmonary tuberculosis patients using propensity score matching

Feng Zhang, Yong Gao, Tuantuan Li, Wei Zhang●*

No.2 People's Hospital of Fuyang City, Fuyang Infectious Disease Clinical College of Anhui Medical University, Fuyang, Anhui, China

* 94499420@qq.com

## Abstract

### Objective

This study aims to explore factors influencing false-negative results in Interferon-Gamma Release Assay (IGRA) for patients with Pulmonary Tuberculosis (PTB), and develop a nomogram model to predict IGRA false negatives, thereby optimizing clinical diagnosis and treatment decisions.

### Methods

Data were collected from January 2023 to September 2024 at the Second People's Hospital of Fuyang City, involving 143 PTB patients. Among them, 63 patients who were IGRA negative but pathogen positive formed the observation group, while 80 patients who were both IGRA and pathogen positive constituted the control group. Propensity Score Matching (PSM) was used to balance potential confounding factors between the two groups. Clinical characteristics and laboratory indicators were compared, followed by logistic regression analysis to identify independent risk factors affecting IGRA results. A nomogram model was constructed based on these factors and its predictive performance evaluated.

### Results

After PSM, each group consisted of 55 patients. The observation group showed significantly lower levels of white blood cell count (WBC), neutrophil count (NEUT), lymphocyte count (LYM), red blood cell count (RBC), hemoglobin (HGB), and albumin (ALB) compared to the control group (P < 0.05). Multivariate analysis ultimately identified RBC, ALB and NLR as independent predictors of IGRA false-negativity. The developed nomogram model demonstrated good calibration ($\chi^2$ = 4.482, P = 0.811), with an area under the receiver operating characteristic curve (AUC) of 0.764 (95%

**Data availability statement:** All relevant data are within the paper and its Supporting Information files.

**Funding:** This work was supported by Scientific Research Project of Fuyang Municipal Health Commission (FY2023-036).

**Competing interests:** The authors have declared that no competing interests exist.

CI: 0.675−0.853). Decision curve analysis indicated that the net benefit of predicting false-negative IGRA results using this nomogram model was greater than 0 when the threshold probability ranged from 0.15 to 0.95.

## Conclusion

Decreased RBC/ALB and elevated NLR may be pivotal factors contributing to false-negative IGRA results in PTB patients. The three-variable nomogram shows enhanced predictive performance, serving as a quantitative tool to identify high-risk cases, particularly for patients with malnutrition or pronounced inflammatory status.

---

## Introduction

Pulmonary Tuberculosis (PTB), caused by *Mycobacterium tuberculosis*, is a chronic infectious disease that poses a significant threat to global public health and remains one of the leading causes of death from infectious diseases worldwide [1,2]. According to the latest report from the World Health Organization (WHO), millions of people worldwide are infected with and die from PTB each year, with particularly high incidence and mortality rates in developing countries. Notably, China is among the high-burden countries for PTB, ranking third globally in terms of incidence. In China, the estimated number of PTB cases is 748,000, resulting in an incidence rate of 52 per 100,000 population [3]. Early diagnosis plays a vital role in controlling the transmission of PTB and is essential for improving patient outcomes. Timely detection not only aids in the prompt initiation of appropriate treatment but also helps prevent the further spread of the disease within communities [4].

Interferon-Gamma Release Assays (IGRAs) represent a novel in vitro immunological testing method primarily recommended for latent tuberculosis infection (LTBI) screening rather than active PTB diagnosis in adults. However, in clinical practice, IGRA tests are sometimes inadvertently used during initial diagnostic workups for suspected PTB cases before microbiological confirmation is available. This is due to their high specificity and the advantage of not being affected by Bacille Calmette-Guérin (BCG) vaccination [5,6]. Although IGRAs demonstrate high specificity in detecting latent and active TB infections, their sensitivity remains suboptimal in clinical practice. Recent studies suggest that false-negative IGRA results may arise from impaired cellular immunity, such as in patients with malnutrition, immunosuppressive conditions, or early-stage acute TB infections [7–9]. These limitations highlight the importance of understanding factors contributing to false-negative IGRA results, particularly in settings where the test may be misapplied to active TB diagnosis. The clinical relevance of investigating IGRA false negativity in PTB patients lies in two key aspects:To improve interpretation of LTBI screening results in populations with high PTB prevalence, where distinguishing between latent and active infection is challenging;To inform clinical judgment when IGRA is unexpectedly negative in high-risk patients undergoing evaluation for possible TB.

Previous studies on IGRA false negativity predominantly relied on multivariate regression, which may inadequately address confounding biases. Unlike randomized trials where treatment assignment is controlled, observational studies require methods like PSM to simulate randomization by balancing measured confounders [10]. Here, PSM serves as a statistical tool to reduce selection bias, not as an intervention itself. This study aims to systematically analyze hematological and biochemical indicators associated with false-negative IGRA results in culture-confirmed PTB patients through PSM, and develop a nomogram model to quantify their combined predictive value for identifying false-negative IGRA readings rather than for PTB diagnosis.

## Materials and methods

### Study subjects

This study adopted a retrospective cohort design, recruiting a total of 143 patients diagnosed with PTB at the Second People's Hospital of Fuyang City, Anhui Province, from January 2023 to September 2024. Patients were categorized into two groups based on their IGRA test results: the observation group comprised 63 patients who tested IGRA negative but were pathogen positive, whereas the control group included 80 patients who tested both IGRA positive and pathogen positive. This approach allowed for a detailed comparison between patients with discordant and concordant IGRA and pathogen test results. The study protocol received approval from the Ethics Committee of the Second People's Hospital of Fuyang City (Approval No: 20211231014) and complied with both the Helsinki Declaration. All participants provided written informed consent prior to their inclusion in this study. To ensure patient confidentiality, all personal identifiers (including names, ID numbers and admission dates) were permanently removed from the dataset prior to analysis, with patients assigned unique study codes. Only aggregated data are presented in this study to ensure complete protection of participant privacy.

### Inclusion and exclusion criteria

Inclusion Criteria: (1) Patients diagnosed with PTB according to the "WS288—2017 Pulmonary Tuberculosis Diagnosis" guidelines [11]; (2) All included patients had a confirmed positive pathogen status verified by the Xpert MTB/RIF assay and simultaneously underwent IGRA testing; (3) Patients with complete clinical records and laboratory test data. Exclusion Criteria: (1)Autoimmune diseases, malignancies, organ transplants, and HIV positivity, among other conditions that may affect immune function; (2)Patients who had received immunosuppressive therapy or used steroid medications. Rationale for Exclusion:The use of immunosuppressive therapies and steroids can significantly alter immune responses, potentially leading to false-negative IGRA results. These treatments suppress the body's ability to mount an effective immune response, which is crucial for the accurate detection of Mycobacterium tuberculosis-specific T-cell responses in IGRA tests. Corticosteroids are known to impair lymphocyte proliferation and cytokine production, further compromising the reliability of IGRA outcomes [12]. By excluding these patients, we aimed to ensure a more homogeneous study population and minimize confounding factors related to altered immune states.

The overview of the design process in this study, which details the main steps from patient screening to data analysis (Fig 1).

### Patients data collection

The clinical data collected for hospitalized patients included demographic information such as gender, age, and body mass index (BMI), as well as a range of symptoms including cough, sputum production, hemoptysis, dyspnea, loss of appetite, and fever. Medical history details were also gathered, covering smoking and alcohol consumption history, the presence of diabetes mellitus and hypertension, chronic lung diseases such as chronic obstructive pulmonary disease, emphysema, and bronchiectasis, and the presence of pulmonary cavitation. Body mass index (BMI) was calculated as weight (kg) divided by height squared (m²).

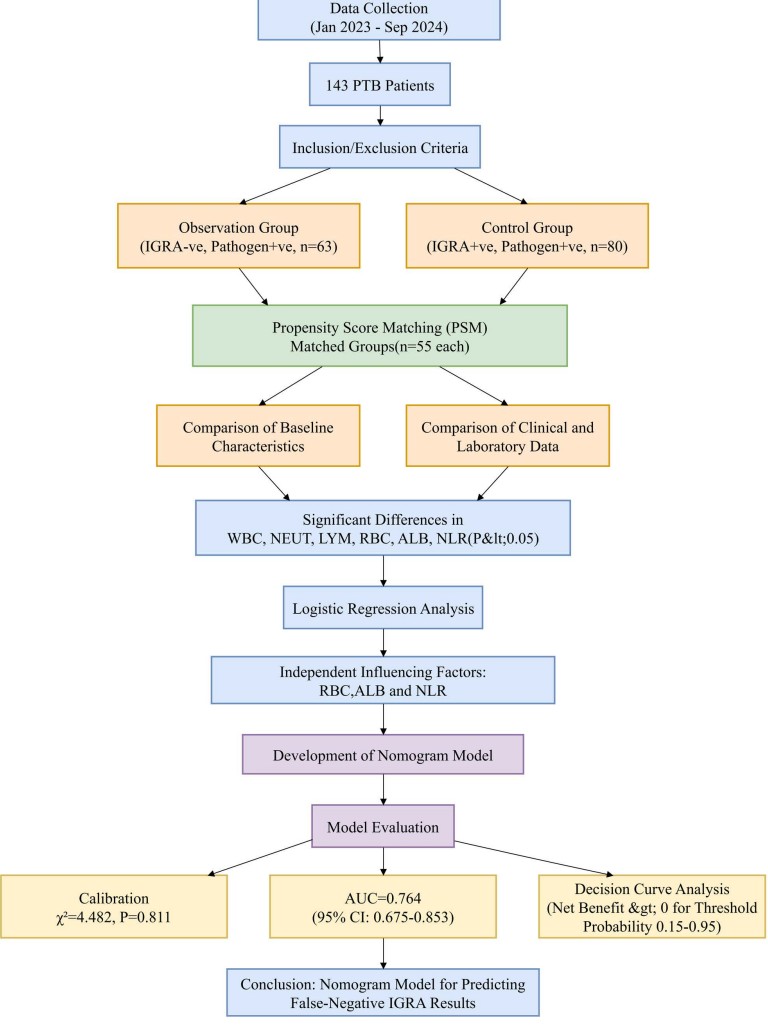

**Fig 1. Flowchart of Study Design and Analysis.**

## Clinical laboratory data

For the blood tests, 5 mL of whole blood was collected in BD vacuum tubes with lithium heparin anticoagulant. One milliliter of blood was added to each of the N, P, and T reaction tubes, which were then incubated at a constant temperature of 37°C for 24 hours. Following incubation, IGRA testing was performed using the Wantai CARIS200 chemiluminescent immunoassay analyzer and its corresponding reagents.2 mL of blood containing EDTA-K2 anticoagulant was drawn and analyzed on a SYSMEX XE2100 automated hematology analyzer along with its corresponding reagent kit to measure white blood cell count (WBC), neutrophil count (NEUT), lymphocyte count (LYM), red blood cell count (RBC), hemoglobin (HGB), and platelet count (PLT).3 mL of fasting morning blood was collected from patients and analyzed on a Hitachi 7600 automated biochemical analyzer to determine albumin (ALB) and C-reactive protein (CRP) levels, The assay kits are from Zhongsheng North Control Biotechnology Co. Based on the collected data, ratios including the Neutrophil-to-Lymphocyte Ratio (NLR) and Platelet-to-Lymphocyte Ratio (PLR) were computed.

For the Xpert MTB/RIF test, 1–2 mL of the specimen was transferred to a 15 mL centrifuge tube. Depending on the nature of the specimen, 1–2 times the volume of Sample Reagent (SR) was added. The mixture was vortexed for 30 seconds until no clumps remained, then left to stand for 20 minutes, during which it was shaken at least twice. After this period, 2 mL of the thoroughly liquefied specimen was aspirated and transferred to the Xpert MTB/RIF cartridge for analysis [13].

## Statistical analysis

Statistical analyses were performed using SPSS version 26.0. For continuous data, the Kolmogorov-Smirnov (K-S) test was used to assess normality. Data conforming to a normal distribution were analyzed using independent samples t-tests and expressed as mean ± standard deviation (SD). Data with skewed distributions were analyzed using the Mann-Whitney U test and presented as median (M) with interquartile range [P25, P75]. Categorical data were summarized as counts (n) and percentages (%) and analyzed using the chi-square ($\chi^2$) test. To address potential confounding by clinical presentation (e.g., symptoms/comorbidities), PSM was conducted using a 1:1 nearest neighbor method with a caliper value of 0.02 on demographic and clinical variables [14] (Table 1). Comorbidities (e.g., diabetes, hypertension), behavioral factors (e.g., smoking, alcohol use), and laboratory parameters were intentionally excluded from matching to preserve their natural variance for subsequent analysis. This approach aligns with our study's aim to evaluate unadjusted associations between these factors and IGRA false negativity, as they may mediate rather than confound the biological pathways of immune suppression [15], while also enabling their direct incorporation into the nomogram construction. The dependent variable for matching was whether the IGRA result was false negative, while the covariates included gender, age, BMI, cough, sputum production, hemoptysis, dyspnea, loss of appetite, and fever. Logistic regression analysis was employed to identify independent risk factors for false-negative IGRA results. The complete output of the propensity score estimation model is provided in S2 Table. Stepwise regression (entry α = 0.05, removal α = 0.10) was used for multivariate logistic regression to determine final predictors. The nomogram prediction model was constructed using R version 4.4.2 with the "rms" package. Calibration curves, receiver operating characteristic (ROC) curves, and Hosmer-Lemeshow (HL) goodness-of-fit tests were utilized to evaluate the accuracy of the nomogram model using the "pROC" and "rms" packages. Decision curve analysis (DCA) was conducted using the "rmda" package to assess the clinical utility of the prediction model. A p-value less than 0.05 (*P* < 0.05) was considered to indicate statistical significance.

Table 1. Comparison of baseline characteristics before and after propensity matching between the two groups.

| Characteristic | Before propensity score matching | | | After propensity score matching | | |
|---|---|---|---|---|---|---|
| | Observation group(n = 63) | Control group(n = 80) | P | Observation group(n = 55) | Control group(n = 55) | P |
| Age[M(P25, P75)] | 67.56 ± 14.62 | 61.04 ± 19.41 | 0.028 | 65.84 ± 14.74 | 65.11 ± 16.70 | 0.809 |
| Gender[n(%)] | | | 0.653 | | | 0.621 |
| Male | 53(84.13) | 65(81.25) | | 46(83.64) | 44(80.00) | |
| Female | 10(15.87) | 15(18.75) | | 9(16.36) | 11(20.00) | |
| BMI(mean±SD) | 19.80 ± 3.10 | 20.44 ± 3.02 | 0.216 | 20.01 ± 3.24 | 20.38 ± 3.11 | 0.534 |
| Coughing[n(%)] | 56(88.89) | 69(86.25) | 0.637 | 49(89.09) | 48(87.27) | 0.768 |
| Sputum production[n(%)] | 55(87.30) | 65(81.25) | 0.328 | 48(87.27) | 48(87.27) | 1.000 |
| Hemoptysis[n(%)] | 5(7.94) | 8(10.00) | 0.670 | 4(7.27) | 2(3.64) | 0.401 |
| Dyspnea[n(%)] | 30(47.62) | 28(35.00) | 0.127 | 24(43.64) | 26(47.27) | 0.702 |
| Loss of appetite[n(%)] | 26(41.27) | 22(27.50) | 0.083 | 18(32.73) | 18(32.73) | 1.000 |
| Fever[n(%)] | 30(47.62) | 35(43.75) | 0.645 | 27(49.09) | 25(45.45) | 0.702 |

## Results

### Comparison of baseline characteristics before and after matching between the two groups

Before propensity score matching, a statistically significant difference in age was observed between the two groups (*P*<0.05; see S3 Table for full unmatched cohort characteristics). After successfully matching 55 pairs of patients using PSM, no statistically significant differences were found between the groups for any of the factors (*P*>0.05) (Table 1). The propensity score distribution plot illustrates that the distribution was imbalanced between the two groups before matching but became markedly more consistent after matching, with differences notably reduced. Standardized mean differences (SMD) indicated that most variables achieved acceptable balance (SMD<0.1; detailed balance metrics are shown in S1 Table), with only minor imbalance remaining in hemoptysis (SMD=0.15) (Fig 2).

### Comparative analysis of related factors between the two groups after matching

After matching, a comparative analysis was conducted on smoking history, alcohol consumption history, diabetes mellitus, hypertension, chronic lung diseases, pulmonary cavitation, and laboratory indicators between the two groups. The results showed significant differences in WBC, NEUT, LYM, RBC, ALB, and NLR (*P*<0.05). No statistically significant differences were found in the other factors (*P*>0.05) (Table 2). The matched cohort laboratory and clinical data are comprehensively summarized in S4 Table.

### Multivariate binary logistic regression analysis

Preliminary multivariate logistic regression with IGRA false-negativity as the dependent variable and WBC, NEUT, LYM, RBC, ALB, NLR as covariates (Table 3) identified RBC and ALB as independent factors (P<0.05). Stepwise regression refinement ultimately confirmed RBC, ALB and NLR as significant predictors in the final model (Table 4).

### Development and validation of the nomogram model

A risk prediction nomogram model for IGRA false-negative results in pulmonary tuberculosis patients was constructed using RBC, ALB, and NLR as predictors (Fig 3). The Hosmer-Lemeshow goodness-of-fit test demonstrated satisfactory calibration of the nomogram model (χ²=4.482, *P*=0.811) (Fig 4). ROC curve analysis revealed an AUC of 0.764 (95% CI:

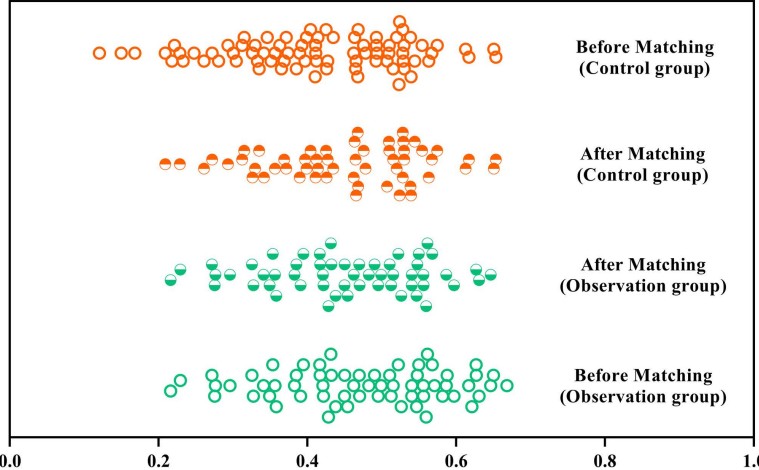

**Fig 2. Distribution of scores before and after matching in the two groups.**

**Table 2. Comparative analysis of related factors between the two groups after matching.**

| Variables | Observation group(n = 55) | Control group(n = 55) | Statistical value | P value |
|---|---|---|---|---|
| Smoking history[n(%)] | 28(50.91) | 26(47.27) | 0.146 | 0.702 |
| Alcohol consumption history[n(%)] | 22(40.00) | 24(43.64) | 0.149 | 0.699 |
| Diabetes[n(%)] | 15(27.27) | 12(21.82) | 0.442 | 0.506 |
| Hypertension[n(%)] | 10(18.18) | 11(20.00) | 0.059 | 0.808 |
| Chronic lung disease[n(%)] | 15(27.27) | 9(16.36) | 1.919 | 0.166 |
| Pulmonary cavitation[n(%)] | 22(40.00) | 20(36.36) | 0.154 | 0.695 |
| WBC[M(P25, P75)] | 8.53(6.28,10.04) | 6.47(5.88,7.94) | −2.923 | 0.003 |
| NEUT[M(P25, P75)] | 5.91(4.05,8.50) | 4.50(3.34,5.74) | −3.252 | 0.001 |
| LYM(mean±SD) | 1.12±0.59 | 1.36±0.64 | −2.029 | 0.045 |
| RBC(mean±SD) | 3.71±0.66 | 4.00±0.66 | −2.349 | 0.021 |
| HGB[M(P25, P75)] | 118.00(103.00,129.00) | 117.00(102.00,129.00) | −0.469 | 0.639 |
| PLT[M(P25, P75)] | 267.00(200.00,365.00) | 269.00(191.00,330.00) | −0.499 | 0.618 |
| ALB(mean±SD) | 32.08±3.92 | 35.04±5.46 | −3.263 | 0.001 |
| CRP[M(P25, P75)] | 64.50(10.00,153.60) | 40.20(9.70,84.30) | −1.713 | 0.087 |
| NLR[M(P25, P75)] | 5.89(2.66,11.11) | 3.60(2.49,5.83) | −2.867 | 0.004 |
| PLR[M(P25, P75)] | 247.76(161.86,424.49) | 214.41(140.41,335.06) | −1.767 | 0.077 |

**Table 3. Multivariate Binary Logistic Regression Analysis.**

| Variable | β | SE | Wald χ² | P value | OR | 95%CI | |
|---|---|---|---|---|---|---|---|
| | | | | | | lower limit | upper limit |
| WBC | 0.102 | 0.350 | 0.085 | 0.771 | 1.107 | 0.558 | 2.198 |
| NEUT | −0.094 | 0.421 | 0.050 | 0.823 | 0.910 | 0.399 | 2.075 |
| LYM | 0.365 | 0.640 | 0.325 | 0.568 | 1.441 | 0.411 | 5.051 |
| RBC | −0.892 | 0.393 | 5.160 | 0.023 | 0.410 | 0.190 | 0.885 |
| ALB | −0.127 | 0.047 | 7.101 | 0.008 | 0.881 | 0.803 | 0.967 |
| NLR | 0.199 | 0.112 | 3.181 | 0.075 | 1.220 | 0.980 | 1.518 |

**Table 4. Final Multivariate Logistic Regression Model (Stepwise Selection).**

| Variable | β | SE | Wald χ² | P value | OR | 95%CI | |
|---|---|---|---|---|---|---|---|
| | | | | | | lower limit | upper limit |
| RBC | −0.880 | 0.388 | 5.146 | 0.023 | 0.415 | 0.194 | 0.887 |
| ALB | −0.120 | 0.047 | 6.655 | 0.010 | 0.887 | 0.809 | 0.972 |
| NLR | 0.161 | 0.055 | 8.669 | 0.003 | 1.174 | 1.055 | 1.307 |

0.675–0.853), with an overall accuracy of 0.709 (95% CI: 0.615–0.792), sensitivity of 0.782 (95% CI: 0.673–0.891), specificity of 0.636 (95% CI: 0.509–0.763), positive predictive value (PPV) of 0.683 (95% CI: 0.568–0.797), and negative predictive value (NPV) of 0.745 (95% CI: 0.620–0.869) (Fig 5). Decision curve analysis showed that the net benefit of using this nomogram to predict IGRA false-negativity exceeded zero when the threshold probability ranged from 0.15 to 0.95 (Fig 6).

## Discussion

While our findings identify predictors of false-negative IGRA results in culture-confirmed PTB patients, they should not be construed as supporting IGRA use for active TB diagnosis. Clinicians should always prioritize microbiological evidence

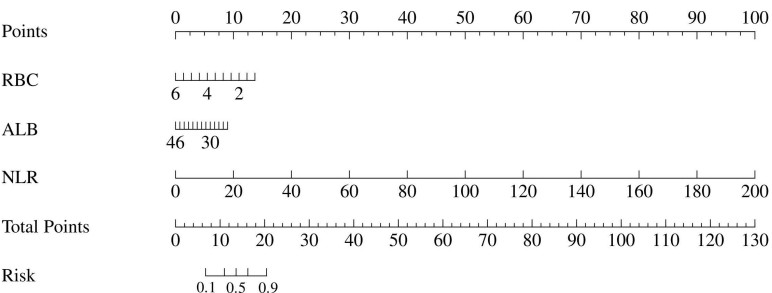

**Fig 3. Nomogram model.**

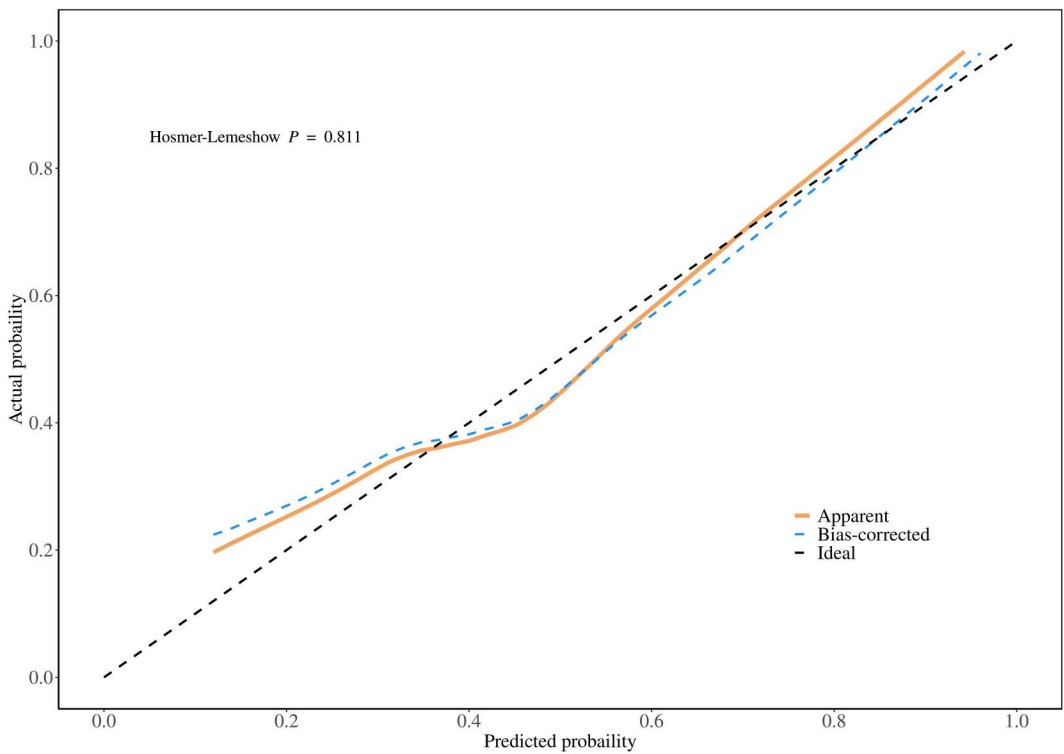

**Fig 4. Calibration curve.**

per WHO guidelines. The practical value of this study lies in helping interpret IGRA results when: (a) screening for LTBI in high-burden settings, and (b) evaluating test reliability in immunocompromised populations.

To address potential confounding biases, we employed PSM – a robust approach that creates comparable groups by balancing observed covariates like age, gender and BMI [16,17]. Unlike traditional multivariate regression, PSM mimics randomized controlled trial conditions by matching participants with similar propensity scores [18]. Our two-stage analytical approach first balanced clinical confounders through PSM (intentionally excluding laboratory variables), then modeled laboratory predictors. This design avoided adjusting for variables potentially on the causal pathway between nutritional status and immune response, which could introduce collider bias as described in causal inference frameworks [19]. The approach enabled direct assessment of hematological factors on IGRA false-negativity while preserving their natural variability for prediction modeling.

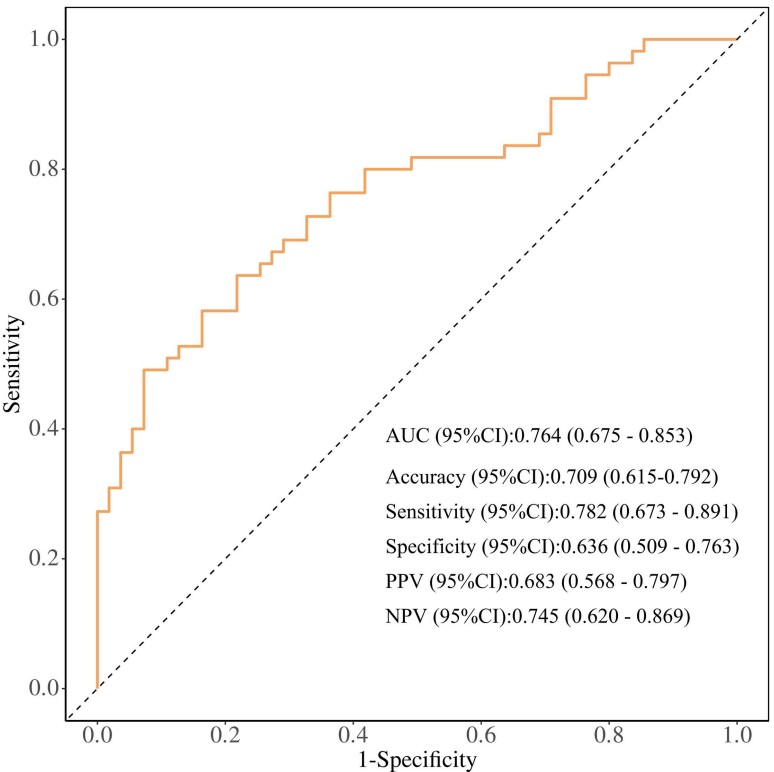

**Fig 5. ROC curve.**

The post-PSM analysis revealed a distinct immunological profile in patients with false-negative IGRA results. We observed significantly elevated levels of inflammatory markers including NLR accompanied by reduced lymphocyte counts and nutritional parameters (RBC, ALB) in the observation group. This paradoxical combination suggests a dissociation between systemic inflammatory activation and antigen-specific immune responses. The significant association of elevated NLR with false-negative results specifically indicates that neutrophilia-driven inflammation concurrent with lymphopenia may directly inhibit Mycobacterium tuberculosis-specific T cell responses, as demonstrated by the inverse correlation between NLR and IFN-γ production [20]. This aligns with recent findings that excessive inflammatory responses can suppress cellular immunity pathways [21], while malnutrition (reflected by RBC/ALB) independently compromises immune cell function. These hematological and nutritional parameters may serve as more reliable predictors of IGRA performance than traditional risk factors in our matched cohort, as evidenced by the lack of significant differences in comorbidities like diabetes or chronic lung disease [22,23]. However, this does not negate the established impact of such conditions on TB immunity, but rather highlights the need for comprehensive patient assessment incorporating both conventional risk factors and novel hematological indicators.

The multivariate analysis robustly identified RBC, ALB and NLR as independent determinants of false-negative IGRA outcomes, reinforcing the critical association between both nutritional status and inflammatory imbalance with cellular immune competence. NLR's predictive value persisted after adjusting for nutritional markers, suggesting inflammation-mediated immune suppression may operate through pathways distinct from malnutrition-related mechanisms. These findings corroborate established pathophysiological mechanisms wherein anemia compromises oxygen delivery to immune cells, while hypoalbuminemia reflects both nutritional deficiency and systemic inflammation – collectively impairing

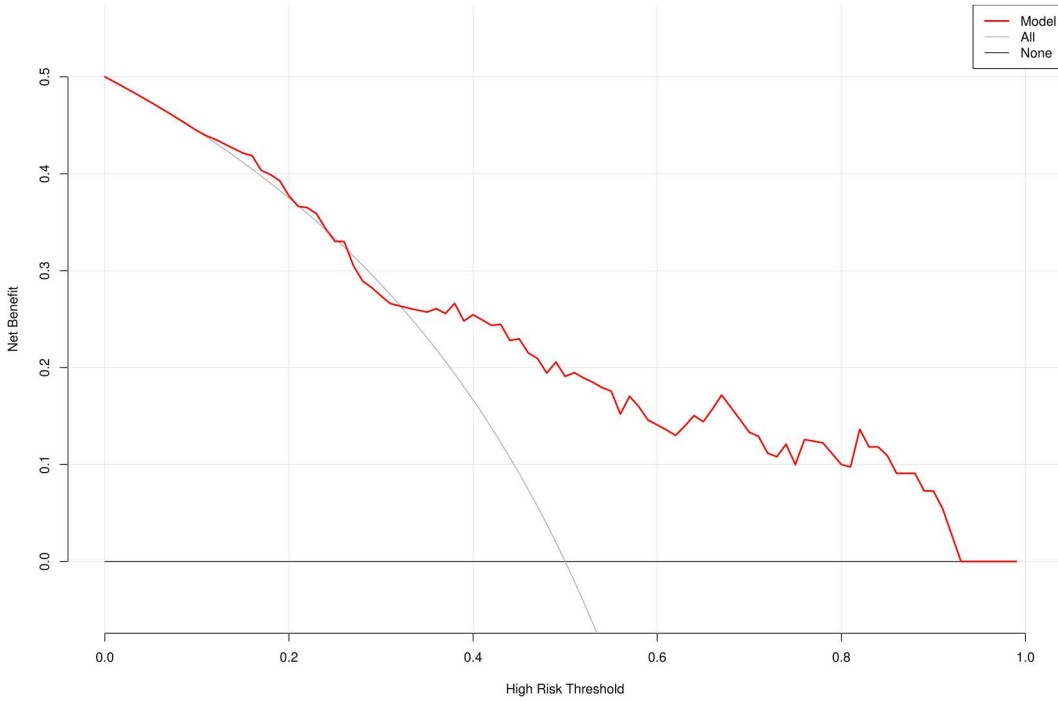

**Fig 6. Decision curves.**

lymphocyte function and IFN-γ production capacity [24]. This biological plausibility is further substantiated by congruent findings from Wang et al. [25] in pediatric TB cohorts and Sharninghausen et al. [26] in migrant populations, collectively suggesting that nutritional markers may serve as universal modifiers of IGRA performance across diverse demographic groups. The consistency of these observations across studies underscores the necessity of contextualizing IGRA interpretations within patients' nutritional frameworks, particularly in resource-limited settings where malnutrition prevalence parallels TB burden. This predictive model advances clinical practice by systematically addressing key challenges in IGRA interpretation. Using readily available laboratory markers improves its practical utility for identifying false-negative risks within existing testing workflows. The model demonstrates consistent performance across diverse clinical scenarios, offering reliable identification of cases requiring confirmatory testing. Its novel integration of nutritional biomarkers provides critical insights often missed in conventional IGRA assessment, particularly the impact of impaired nutritional status on cellular immune responses.

Our findings carry important implications for TB diagnostic protocols: Routine measurement of nutritional markers (hemoglobin, albumin, and iron profiles) should be integrated when interpreting IGRA results in malnourished populations, as low iron levels with elevated NLR may reflect inflammation-induced test impairment requiring confirmatory microbiological testing (e.g., Xpert MTB/RIF). In high-burden settings, the nomogram variables (RBC, ALB, NLR) could optimize resource use through two-stage screening—initial IGRA followed by targeted molecular testing for high-risk cases, aligning with WHO guidelines on active TB diagnosis [27]. While nutritional interventions (e.g., iron supplementation) may enhance immune function, their impact on IGRA accuracy needs prospective validation, emphasizing the need to differentiate nutritional deficiency from inflammation-mediated anemia in clinical decision-making.

## Limitations

Although Propensity Score Matching (PSM) effectively balanced most covariates (all SMDs < 0.1), minor residual imbalances persisted in sex, hemoptysis, and fever. The reversed direction of hemoptysis difference post-matching likely reflects stochastic variation in this low-prevalence variable. These observations highlight the importance of reporting both statistical and clinical significance of post-matching imbalances. Future studies may benefit from exact matching for rare variables or machine learning-based approaches (e.g., genetic matching) to further improve balance.

As a retrospective study, there may be unidentifiable or uncontrolled confounding factors that affect the generalizability and accuracy of the results. The relatively limited sample size and single-center origin may also restrict the applicability of the study conclusions to broader populations.

Another limitation relates to the clinical application of our nomogram model. Although it demonstrated good calibration and predictive performance, its use requires albumin measurement—a test not routinely performed in the initial evaluation of suspected pulmonary tuberculosis, particularly in outpatient or resource-limited settings. While albumin's association with immune status makes it a valuable predictor, its limited availability should be considered when interpreting or applying the model in practice.

Furthermore, while RBC, ALB, and NLR were identified as predictors of IGRA false-negativity, the causal relationship between these factors and test results remains to be validated in prospective studies. Future research should consider incorporating a wider range of biomarkers and clinical characteristics to comprehensively evaluate their impact on IGRA outcomes.

## Conclusion

In conclusion, this study employed PSM to balance potential confounding factors between groups, thereby exploring the factors influencing false-negative IGRA results in PTB patients. The nomogram model incorporating RBC, ALB and NLR demonstrated robust predictive performance, particularly valuable for patients with malnutrition or pronounced inflammatory status. This tool provides a quantitative approach to identify high-risk false-negative cases, while establishing a novel methodological framework that integrates both nutritional and inflammatory pathways for future research.

## Supporting information

**S1 Table. Balance Check for PSM Variables: Equal vs. Weighted Matching.**
(XLSX)

**S2 Table. Propensity Score Estimation Model Output: Results from Logistic Regression Analysis.**
(XLSX)

**S3 Table. Unmatched data.**
(XLSX)

**S4 Table. Matched data.**
(XLSX)

## Author contributions

**Conceptualization:** Feng Zhang.

**Data curation:** Feng Zhang.

**Formal analysis:** Feng Zhang.

**Funding acquisition:** Yong Gao.

**Investigation:** Yong Gao.

**Methodology:** Yong Gao.

**Project administration:** Tuantuan Li.

**Resources:** Tuantuan Li.

**Software:** Tuantuan Li.

**Supervision:** Wei Zhang.

**Validation:** Wei Zhang.

**Visualization:** Wei Zhang.

**Writing – original draft:** Feng Zhang.

**Writing – review & editing:** Wei Zhang.

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
