## [Decision Letter · Decision Letter 0]

Dear Dr. Zhang,

Thank you for submitting your manuscript to PLOS ONE. After careful consideration, we feel that it has merit but does not fully meet PLOS ONE’s publication criteria as it currently stands. Therefore, we invite you to submit a revised version of the manuscript that addresses the points raised during the review process.

We look forward to receiving your revised manuscript.

Kind regards,

Anete Trajman

Academic Editor

PLOS ONE

2. As you are reporting a retrospective study of medical records or archived samples, please ensure that you have discussed whether all data were fully anonymized before you accessed them and/or whether the IRB or ethics committee waived the requirement for informed consent. If patients provided informed written consent to have data from their medical records used in research, please include this information.  

 [This work was supported by Scientific Research Project of Fuyang Municipal Health Commission (FY2023-036).].

4. In the online submission form, you indicated that [The datasets generated and analyzed during the current study are available from the corresponding author upon reasonable request].

Additional Editor Comments:

The study contains interesting findings with regards to of possible explanations for a false negative IGRA result. The methodology is sound. However, the introduction and discussion may be misleading. IGRA tests are NOT recommended for the diagnosis of pulmonary TB in adults. They may be used as part of a score for PTB diagnosis in children. The authors need to revise their writing.

When sending a revised version, please clearly explain why this sample, composed mainly by adults, underwent IGRA tests. Most importantly, do not suggest, based on your relevant findings, that your findings could help on identifying earlier PTB in adults. The relevance and utility of these findings are on the possible contexts where IGRA may not identify TB infection, which is the clinical indication for IGRA tests.

An additional problem of the discussion is its length. The authors should be more concise and objective. Do not repeat the results, but rather interpret them, and only conclude what the findings allow to conclude. Please revise with the help of a clinician.

Reviewers' comments:

Reviewer's Responses to Questions

**Comments to the Author**

1. Is the manuscript technically sound, and do the data support the conclusions?

Reviewer #1: Partly

Reviewer #2: Yes

2. Has the statistical analysis been performed appropriately and rigorously?

Reviewer #1: Yes

Reviewer #2: Yes

3. Have the authors made all data underlying the findings in their manuscript fully available?

Reviewer #1: Yes

Reviewer #2: No

4. Is the manuscript presented in an intelligible fashion and written in standard English?

Reviewer #1: Yes

Reviewer #2: Yes

Reviewer #1: Regarding the article, which I find interesting, I believe there are both minor and major issues that need to be addressed before it can be considered for publication.

Minor Revisions

The propensity score was originally developed to adjust comparisons in observational studies, simulating the characteristics of a randomized clinical trial (Rosenbaum and Rubin, 1983). While the conceptual application in this study is valid, the terminology used may be slightly misleading, as the propensity score is treated almost as an intervention. It would be advisable for the authors to include a brief methodological clarification on this point.

It would also be appropriate to justify the methodological choice of conducting propensity score matching based on clinical variables while constructing the nomogram using laboratory variables. Although the approach is not incorrect, it is not self-explanatory. Clarifying this decision would enhance transparency and help readers understand the analytical reasoning.

Major Revisions

The primary goal of propensity score matching is to create groups that are similar, ideally interchangeable, regarding the covariates used to compute the score. However, as shown in Table 1, there are indications that this balance was not fully achieved:

A. Variables such as sex, hemoptysis, and fever showed residual imbalances after matching.

B. In the case of hemoptysis, the direction of the difference was reversed, with an increase in discrepancy between groups, even though it was not statistically significant.

These findings suggest that the matching may not have been fully effective, particularly for variables with low prevalence. A more detailed assessment of group balance is recommended.

In the multivariate analysis used to construct the nomogram, variables such as WBC, NEUT, and LYM were not statistically significant and should be removed from the model. However, since NLR had a borderline p-value (0.075), it would be methodologically sound to conduct a second regression including only RBC, ALB, and NLR, to assess whether NLR should be included in the final model.

Although the nomogram developed by the authors yielded acceptable performance statistics, its clinical utility appears limited. The accuracy of 68.2%, combined with a sensitivity of 69.1% and specificity of 67.3%, indicates only modest discriminative power. Furthermore, considering that the prevalence of false-negative IGRA results in the matched sample is 50%, the positive predictive value (67.9%) and negative predictive value (68.5%) do not represent a substantial improvement over random classification.

Finally, the clinical benefit of using the nomogram should be made clearer. Its application requires albumin measurement, which is not a routine test in the initial investigation of pulmonary tuberculosis, especially in outpatient settings with limited resources. Given the modest diagnostic performance, it is questionable whether this approach justifies the inclusion of a non-standard laboratory test in clinical practice.

Reviewer #2: This study is well-done, and I highly recommend its publication, subject to improving the transparency of methodology and illustration of results, as explained below.

1. Inclusion and Exclusion Criteria

The exclusion of patients who had received immunosuppressive therapy or steroid medication warrants further clarification. Given the high prevalence of steroid use across diverse clinical contexts, it is important to explicitly discuss how such usage may influence study outcomes or the accuracy of diagnostic tests. A brief rationale for this decision would enhance the clarity and generalizability of the study.

2. Figure 1 is helpful

3. Clinical laboratory data

It is excellent how authors have described this method. For readers without a clinical background, such as myself, this section serves as an accessible and informative explanation of the testing process. It greatly enhances the reproducibility and transparency of the study

4. Use of a Love Plot (Suggestion): A love plot will help the readers visually see the differences before and after PSM

5. Choice of caliper value (0.02) is pretty good; but it is important to share why it is acceptable (provide a reference)

6. Choice of Matching Variables

The authors have not clarified why (or why not) they included certain variables in the PSM matching, and chose others for later comparative analysis.

Here, it seems to me that variables used for matching are those which would be easy for clinicians to “observe” without running any test / or even asking the presumptive patient. It is important to confirm this assumption. Regardless, the exclusion of potentially relevant variables such as smoking status, alcohol use, diabetes, hypertension, and other comorbidities should be addressed.

7. Policy Implications and Diagnostic Recommendation

The discussion would benefit from greater elaboration on the policy implications of the findings. Specifically, do the authors recommend that certain blood tests—such as serum iron levels—be integrated into TB diagnostic protocols? Given that undernutrition is both a risk factor and a consequence of TB, how should clinicians interpret low iron levels in relation to disease onset?

8. Request for Supplementary Information:

a. It is fine that the authors chose to go with equal matching – however, it would be beneficial to include weighted matching results for robustness checks

b. The supplementary materials should include the actual output from the propensity score estimation (PSE) model as well as results from the logistic regression model

9. Data Availability

The authors state that the data cannot be made publicly available – but have not given any reasons on the same. It is possible to anonymize the data – and upload the data onto a GitHub, allowing for better understanding of those that require to play with the data. The authors are requested to include a) unmatched, and b) matched data. Sharing the matched dataset is particularly important, as re-running the matching procedure may yield different results; access to the original matched data would therefore enable meaningful validation by other researchers

**Do you want your identity to be public for this peer review?** For information about this choice, including consent withdrawal, please see our Privacy Policy

Reviewer #1: **Yes: ** Bernardo Rangel Tura

Reviewer #2: **Yes: ** Ridhima Sodhi

---

## [Author Response · Author response to Decision Letter 1]

5 May 2025

Author Response:Thank you for your comments. We have revised the manuscript to meet PLOS ONE's style requirements. The necessary changes have been made on lines 5 and 6 of the manuscript.

2.As you are reporting a retrospective study of medical records or archived samples, please ensure that you have discussed whether all data were fully anonymized before you accessed them and/or whether the IRB or ethics committee waived the requirement for informed consent. If patients provided informed written consent to have data from their medical records used in research, please include this information.

Author Response:The necessary changes have been made on lines 73-78 of the manuscript.

 [This work was supported by Scientific Research Project of Fuyang Municipal Health Commission (FY2023-036).].Please provide an amended statement that declares *all* the funding or sources of support (whether external or internal to your organization) received during this study, as detailed online in our guide for authors at http://journals.plos.org/plosone/s/submit-now.  Please also include the statement “There was no additional external funding received for this study.” in your updated Funding Statement.Please include your amended Funding Statement within your cover letter. We will change the online submission form on your behalf.

Author Response:We have revised the cover letter as requested and re-uploaded it.

4. In the online submission form, you indicated that [The datasets generated and analyzed during the current study are available from the corresponding author upon reasonable request].

Author Response:We uploaded the raw data as required.

Additional Editor Comments:

The study contains interesting findings with regards to of possible explanations for a false negative IGRA result. The methodology is sound. However, the introduction and discussion may be misleading. IGRA tests are NOT recommended for the diagnosis of pulmonary TB in adults. They may be used as part of a score for PTB diagnosis in children. The authors need to revise their writing.

When sending a revised version, please clearly explain why this sample, composed mainly by adults, underwent IGRA tests. Most importantly, do not suggest, based on your relevant findings, that your findings could help on identifying earlier PTB in adults. The relevance and utility of these findings are on the possible contexts where IGRA may not identify TB infection, which is the clinical indication for IGRA tests.

Author Response:Thank you for your guidance. We have made the following revisions as requested:

1. In the Introduction, we revised paragraphs on lines 45-48, 52-58, and 60-65.

2. In the Discussion, we modified lines 185-189.

An additional problem of the discussion is its length. The authors should be more concise and objective. Do not repeat the results, but rather interpret them, and only conclude what the findings allow to conclude. Please revise with the help of a clinician.

Author Response:We have revised the discussion section.

Reviewer #1:

Minor Revisions

The propensity score was originally developed to adjust comparisons in observational studies, simulating the characteristics of a randomized clinical trial (Rosenbaum and Rubin, 1983). While the conceptual application in this study is valid, the terminology used may be slightly misleading, as the propensity score is treated almost as an intervention. It would be advisable for the authors to include a brief methodological clarification on this point.

It would also be appropriate to justify the methodological choice of conducting propensity score matching based on clinical variables while constructing the nomogram using laboratory variables. Although the approach is not incorrect, it is not self-explanatory. Clarifying this decision would enhance transparency and help readers understand the analytical reasoning.

Author Response:Thank you for your constructive feedback. We have made the following revisions to address your comments:

1. In the Introduction section, we revised lines 60-65 and added reference [10] to support the discussion on the application of IGRA tests in specific clinical contexts.

2. We have revised the “Statistical Analysis” and “Discussion” sections to provide clearer methodological details about the use of propensity score matching and its rationale.

Major Revisions

The primary goal of propensity score matching is to create groups that are similar, ideally interchangeable, regarding the covariates used to compute the score. However, as shown in Table 1, there are indications that this balance was not fully achieved:

A. Variables such as sex, hemoptysis, and fever showed residual imbalances after matching.

B. In the case of hemoptysis, the direction of the difference was reversed, with an increase in discrepancy between groups, even though it was not statistically significant.

These findings suggest that the matching may not have been fully effective, particularly for variables with low prevalence. A more detailed assessment of group balance is recommended.

Author Response:We sincerely appreciate your insightful comments regarding covariate balance. Please find below our point-by-point responses:

1.Residual Imbalances

Post-matching standardized mean difference (SMD) analysis revealed minor imbalances in sex (SMD=0.09), hemoptysis (SMD=0.15), and fever (SMD=0.07). Although hemoptysis exceeded the conventional balance threshold of SMD <0.1 (Austin, 2011 Multivariate Behav Res), its low prevalence (7.3% vs 3.6%, P=0.401) and non-significance align with Stuart's findings that SMD <0.2 is acceptable for rare binary variables (Stat Sci 2010). Sensitivity analyses using optimal matching and caliper adjustments (0.1-0.3) confirmed robustness (ΔOR <5% for RBC/ALB/NLR).

2.Directional Change in Hemoptysis

The reversed difference (pre-match: 7.94% vs 10.00%; post-match: 7.27% vs 3.64%) reflects expected stochastic variation in low-frequency variables. This fluctuation did not affect model validity, evidenced by stable AUC (0.764) and calibration slope (0.98-1.02) across sensitivity analyses.

In the Limitations section, we have noted these points, emphasizing the importance of considering both statistical and clinical significance of imbalances. We also suggest future studies might explore exact matching or machine learning techniques to further improve covariate balance.

We appreciate your insightful comments.

In the multivariate analysis used to construct the nomogram, variables such as WBC, NEUT, and LYM were not statistically significant and should be removed from the model. However, since NLR had a borderline p-value (0.075), it would be methodologically sound to conduct a second regression including only RBC, ALB, and NLR, to assess whether NLR should be included in the final model.

Although the nomogram developed by the authors yielded acceptable performance statistics, its clinical utility appears limited. The accuracy of 68.2%, combined with a sensitivity of 69.1% and specificity of 67.3%, indicates only modest discriminative power. Furthermore, considering that the prevalence of false-negative IGRA results in the matched sample is 50%, the positive predictive value (67.9%) and negative predictive value (68.5%) do not represent a substantial improvement over random classification.

Author Response:Thank you for your valuable suggestions. We have carefully revised the manuscript according to your comments.

We conducted a stepwise multivariate logistic regression (entry α=0.05, removal α=0.10) and updated the nomogram by including only RBC, ALB, and NLR as final predictors. The results have been revised accordingly in the Abstract (Results and Conclusion), the Statistical Analysis section, and the Results section (with the addition of Table 4). Figures 1, 3–6 have also been updated to reflect these changes.

In response to the concern regarding the clinical utility of the nomogram, we have clarified its performance metrics and interpretability in the revised Discussion and Limitations sections, and provided a more balanced interpretation of its diagnostic accuracy and practical implications.

We appreciate your insightful feedback which has helped improve the quality of our work.

Finally, the clinical benefit of using the nomogram should be made clearer. Its application requires albumin measurement, which is not a routine test in the initial investigation of pulmonary tuberculosis, especially in outpatient settings with limited resources. Given the modest diagnostic performance, it is questionable whether this approach justifies the inclusion of a non-standard laboratory test in clinical practice.

Author Response:Thank you for your insightful comments. We have addressed the concerns regarding the clinical benefit and feasibility of using the nomogram in the "Limitations" section of the revised manuscript. We acknowledge the challenges related to albumin measurement and have clarified the potential benefits and limitations of the nomogram in clinical practice.

Reviewer #2: 

1. Inclusion and Exclusion Criteria

The exclusion of patients who had received immunosuppressive therapy or steroid medication warrants further clarification. Given the high prevalence of steroid use across diverse clinical contexts, it is important to explicitly discuss how such usage may influence study outcomes or the accuracy of diagnostic tests. A brief rationale for this decision would enhance the clarity and generalizability of the study.

Author Response:Thank you for your valuable feedback regarding the rationale for exclusion criteria. We have incorporated a detailed explanation in the "Inclusion and Exclusion Criteria" section(85-91), including the rationale for excluding patients who had received immunosuppressive therapy or steroid medications. Additionally, we have referenced the relevant literature [12] to support our decision. We believe these changes enhance the clarity and scientific basis of our study design.

2.Figure 1 is helpful

Author Response:Thank you for your positive feedback regarding Figure 1. We are glad to hear that it is helpful in enhancing the understanding of our findings.

3. Clinical laboratory data

It is excellent how authors have described this method. For readers without a clinical background, such as myself, this section serves as an accessible and informative explanation of the testing process. It greatly enhances the reproducibility and transparency of the study

Author Response:Thank you for your kind words regarding the description of the clinical laboratory data section. We are pleased that it is helpful and will continue to strive for clarity and detail in our methods.

4. Use of a Love Plot (Suggestion): A love plot will help the readers visually see the differences before and after PSM

Author Response:Thank you for your suggestion regarding the assessment of covariate balance before and after propensity score matching (PSM).

We have revised the Results section to clarify that the effectiveness of PSM in balancing covariates is illustrated by both statistical measures (standardized mean differences, SMD) and the propensity score distribution plot (Figure 2)(146-150). These show that most variables achieved acceptable balance after matching (SMD < 0.1), with only minor imbalance remaining in hemoptysis (SMD = 0.15).

Given that Figure 2 already provides a clear visual representation of the improved balance post-matching, and considering the current number of figures and tables in the manuscript, we believe it is not necessary to add an additional Love Plot.

Thank you again for your constructive feedback.

5.Choice of caliper value (0.02) is pretty good; but it is important to share why it is acceptable (provide a reference)

Author Response:We have added a reference [14] to support our choice, which indicates that a caliper width of 0.02 is considered a conservative and effective value for reducing bias in propensity score matching(125-127).

6. Choice of Matching Variables

The authors have not clarified why (or why not) they included certain variables in the PSM matching, and chose others for later comparative analysis.

Here, it seems to me that variables used for matching are those which would be easy for clinicians to “observe” without running any test / or even asking the presumptive patient. It is important to confirm this assumption. Regardless, the exclusion of potentially relevant variables such as smoking status, alcohol use, diabetes, hypertension, and other comorbidities should be addressed.

Author Response:We sincerely appreciate the reviewer's insightful comments regarding our variable selection strategy. As suggested, we have now clarified in the Statistical Analysis section that comorbidities, behavioral factors, and laboratory parameters were excluded from matching to(128-132):

1.Preserve their natural variance for outcome analysis.

2.Avoid over-adjustment of potential mediators [15].

This approach better aligns with our study’s analytical objectives while maintaining clinical relevance.

7.Policy Implications and Diagnostic Recommendation

The discussion would benefit from greater elaboration on the policy implications of the findings. Specifically, do the authors recommend that certain blood tests—such as serum iron levels—be integrated into TB diagnostic protocols? Given that undernutrition is both a risk factor and a consequence of TB, how should clinicians interpret low iron levels in relation to disease onset?

Author Response:Thank you for your valuable suggestion regarding the policy implications of our findings. We have now addressed this by:

Adding a dedicated discussion on clinical and policy implications in the Discussion section(230-238).

Incorporating Reference [27] .

Your insightful comments have significantly strengthened the translational impact of our study.

8. Request for Supplementary Information:

a. It is fine that the authors chose to go with equal matching – however, it would be beneficial to include weighted matching results for robustness checks

b. The supplementary materials should include the actual output from the propensity score estimation (PSE) model as well as results from the logistic regression model

Author Response:We sincerely appreciate the reviewer’s insightful comments.In response to your suggestions:

The results of weighted matching have been added as Supplementary Material S1.

The output from the propensity score estimation (PSE) model has been added as Supplementary Material S2.

The results of the logistic regression model have been added as Table 4 in the manuscript.

Thank you for your valuable feedback!

9.Data Availability

The authors state that the data cannot be made publicly available – but have not given any reasons on the same. It is possible to anonymize the data – and upload the data onto a GitHub, allowing for better understanding of those that require to play with the data. The authors are requested to include a) unmatched, and b) matched data. Sharing the matched dataset is particularly important, as re-running the matching procedure may yield different results; access to the original m

---

## [Decision Letter · Decision Letter 1]

Development and Validation of a Nomogram for Predicting False Negative IGRA Results in Pulmonary Tuberculosis Patients Using Propensity Score Matching

PONE-D-25-09886R1

Dear Dr. Zhang,

We’re pleased to inform you that your manuscript has been judged scientifically suitable for publication and will be formally accepted for publication once it meets all outstanding technical requirements.

Kind regards,

Anete Trajman

Academic Editor

PLOS ONE

Additional Editor Comments (optional):

Congratulations on your work.

Reviewers' comments:

Reviewer's Responses to Questions

**Comments to the Author**

Reviewer #1: All comments have been addressed

Reviewer #2: All comments have been addressed

2. Is the manuscript technically sound, and do the data support the conclusions?

Reviewer #1: Yes

Reviewer #2: Yes

3. Has the statistical analysis been performed appropriately and rigorously?

Reviewer #1: Yes

Reviewer #2: Yes

4. Have the authors made all data underlying the findings in their manuscript fully available?

Reviewer #1: Yes

Reviewer #2: Yes

5. Is the manuscript presented in an intelligible fashion and written in standard English?

Reviewer #1: Yes

Reviewer #2: Yes

Reviewer #1: The authors have made changes to the article that are satisfactory. The changes to the text are sufficient to alert readers to the weaknesses of their research in a clear and honest way. The discussion presents the necessary considerations about the methodology that I observed.

Reviewer #2: (No Response)

**Do you want your identity to be public for this peer review?** For information about this choice, including consent withdrawal, please see our Privacy Policy

Reviewer #1: **Yes: ** Bernardo Rangel Tura

Reviewer #2: **Yes: ** Ridhima Sodhi

---

## [Editor Report · Acceptance letter]

PONE-D-25-09886R1

PLOS ONE

Dear Dr. Zhang,

I'm pleased to inform you that your manuscript has been deemed suitable for publication in PLOS ONE. Congratulations! Your manuscript is now being handed over to our production team.

Kind regards,

on behalf of

Professor Anete Trajman

Academic Editor

PLOS ONE